# IN-CONTEXT ALIGNMENT: RESOLVING REPRESENTATION CONFLICT FOR PARAMETER-EFFICIENT FORGERY DETECTION IN VISION MODELS

## ABSTRACT

Vision Foundation Models (VFMs) have shown remarkable potential in image forensics, yet their content-driven representations often suppress the subtle forensic cues essential for manipulation localization, thereby exhibiting an inherent representation conflict. Conventional full fine-tuning struggles to address this conflict, as it demands extensive parameter updates, risks overfitting, and erodes prior knowledge, leading to poor generalization in diverse forgery detection scenarios. We propose In-Context Alignment (ICA), a parameter-efficient framework that reframes forgery localization as a visual in-context learning task. ICA introduces two complementary prompting mechanisms within frozen VFMs: a Physical-Aware Prompter (PAP) that enhances suppressed low-level forensic signals such as noise and frequency artifacts via a Mixture-of-Experts for adaptive fusion, and a Semantic-Aware Prompter (SAP) that encourages the model to expose semantic inconsistencies in high-level features. With only a small fraction of parameters updated, ICA achieves strong performance across diverse image forgery localization benchmarks and can even compete with fully fine-tuned models. Our results demonstrate that in-context alignment of semantic and forensic representations offers a scalable, robust, and efficient paradigm for advancing visual forensics.

## 1 INTRODUCTION

The rapid advancement of deep generative models, such as GANs Zhu et al. (2025) and Diffusion Models (DDPMs) He et al. (2025), has revolutionized image manipulation, enabling high-fidelity alterations to visual content that challenge the authenticity of digital media. These manipulations, including splicing (merging elements from different images), copy-move (duplicating and repositioning parts within an image), inpainting (contextually filling missing regions), and emerging AI-driven techniques (Figure 1(a)), leave subtle forensic cues, such as physical inconsistencies (e.g., noise disruptions) or semantic dissonances (e.g., irrational object), that are critical for distinguishing tampered from authentic regions, necessitating advanced methods to uncover such traces.

While Vision Foundation Models (VFMs) Siméoni et al. (2025); Kirillov et al. (2023); Xie et al. (2021), pre-trained on massive datasets, excel in semantic understanding, their content-prioritized representations often suppress the low-level cues essential for analyzing manipulated images, resulting in an inherent *representation conflict*. Existing Image Forgery Localization (IFL) methods Cui et al. (2025); Zhu et al. (2024); Guillaro et al. (2023), which rely on specialized architectures to target specific tampering artifacts or optimize feature encoding and fusion process (Figure 1(a)), struggle to leverage VFMs' capabilities effectively. Conventional full fine-tuning of VFMs is inefficient, requiring extensive computational resources, risking catastrophic forgetting of generalizable features, and limiting robustness to novel manipulations under data scarcity. Efforts to address data limitations using large private datasets Guo et al. (2024); Guillaro et al. (2023) are often hindered by inconsistent annotations and limited diversity in real-world post-processing, underscoring the need for a parameter-efficient adaptation paradigm to align semantic and forensic representations.

Addressing these limitations requires a paradigm shift toward efficient VFM alignment, which forms the core motivation of this work: resolving the fundamental machine learning challenge of adapting large-scale VFMs to forgery-sensitive downstream tasks without eroding their general capabilities.

We formalize this intuition as the *Representation Conflict Hypothesis*: VFMs' pre-trained mapping $f_\theta(\cdot)$ (with parameters $\theta$) projects inputs $x$ into a content-oriented representation space $\mathcal{R}_{pre}$ , while ideal forgery detection requires a forgery-sensitive space $\mathcal{R}_{forg}$. The conflict is quantified as the expected divergence: $\mathcal{C} = \mathbb{E}_{x \sim \mathcal{X}}[\Delta(f_\theta(x), f_{forg}(x))]$, where $\Delta$ is a distance metric (e.g., $\ell_2$-norm), and $f_{forg}(\cdot)$ is the oracle forgery detector. This hypothesis extends concepts from transfer learning, where pre-trained representations often require careful adaptation to tasks with conflicting objectives. Full fine-tuning minimizes $\mathcal{C}$ by optimizing all $\theta$ to $\theta'$ via a surrogate loss: $\theta' = \arg\min_{\theta'} \mathbb{E}_{x,y}[\mathcal{L}(f_{\theta'}(x), y)] \approx \mathcal{C}$, assuming $\mathcal{L}$ (the task-specific loss) aligns with the divergence. However, under PAC learning theory Haussler & Warmuth (2018), its generalization error bound is loose: $O\left(\sqrt{\frac{|\theta|}{n}}\right)$, where $n$ is the sample size. For VFMs with massive $|\theta|$ (e.g., billions), this necessitates vast data to converge, increasing risks of overfitting and catastrophic forgetting, where $\theta'$ discards valuable priors encoded in $\theta$.

Our objective is a *minimal-intervention paradigm* that seeks a lightweight prompt function $g(\phi; x)$ (with $|\phi| \ll |\theta|$) to inject forgery-relevant context into the model, yielding aligned representations $f_\theta(x \oplus g(\phi; x))$ that approximate $f_{forg}(x)$. This yields a tighter bound: $O\left(\sqrt{\frac{|\phi|}{n}}\right)$, theoretically justifying prompt learning's data efficiency and robustness. Here we further decompose $\mathcal{C}$ into two approximately orthogonal components, assuming weak correlation between low-level signals and high-level semantics. The physical component $\mathcal{C}_{physical}$ represents the suppression of low-level cues such as noise inconsistencies, whereas the semantic component $\mathcal{C}_{semantic}$ reflects high-level plausibility biases such as semantic dissonances.

To this end, we propose In-Context Alignment (ICA), a parameter-efficient paradigm that reframes forgery localization as a visual in-context learning task. Instead of modifying the backbone, ICA introduces two complementary prompting mechanisms into frozen VFMs without altering $\theta$ (Figure 1(b)). The Physical-Aware Prompter (PAP) explicitly amplifies suppressed forensic cues, such as noise and frequency inconsistencies, and employs a Mixture-of-Experts design to integrate cues adaptively, minimizing $\mathcal{C}_{physical}$. Complementarily, the Semantic-Aware Prompter (SAP) implicitly guides the model to reveal semantic implausibilities embedded within high-level representations, where semantic prompts are decom-

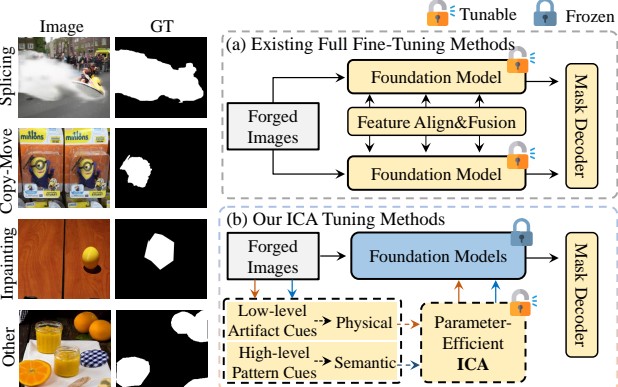

Figure 1: Comparison of tuning paradigms. (a) Existing methods fully fine-tune models to capture limited forgery cues, incurring high computational costs and knowledge loss. (b) ICA enables parameter-efficient tuning by injecting dual-stream physical and semantic prompts into VFMs, preserving prior knowledge and enhancing robustness.

posed into instance-specific and universal cues, minimizing $\mathcal{C}_{semantic}$. By injecting these prompts into intermediate layers, ICA aligns forensic evidence with semantic understanding, thus resolving the representation conflict while reducing data dependency and retaining prior knowledge of VFMs.

Extensive experiments across diverse public benchmarks show that ICA achieves strong performance while tuning only a small fraction of parameters. Without resorting to large-scale fine-tuning or synthetic data, ICA consistently improves robustness against a wide range of manipulations, establishing a new paradigm for efficient and scalable visual forensics. Our contributions are fourfold:

- We introduce the *representation conflict hypothesis* for applying VFMs to forgery localization, and decompose it into *physical* and *semantic* sub-conflicts, providing a principled theoretical lens for understanding why traditional fine-tuning is inefficient.

- We propose *In-Context Alignment (ICA)*, a parameter-efficient framework that integrates a Physical-Aware Prompter and a Semantic-Aware Prompter into frozen VFMs, enabling minimal-intervention alignment of semantic and forensic cues via in-context prompting.

- ICA demonstrates how dual prompting can mitigate semantic-forensic mismatch, revealing the complementarity of explicit physical amplification and implicit semantic guidance.

- Extensive experiments on multiple public benchmarks confirm that ICA achieves efficient adaptation and strong generalization to diverse manipulation types, all while tuning only a small fraction of parameters.

## 2 RELATED WORKS

### 2.1 IMAGE FORGERY LOCALIZATION

Recent progress in image forgery localization (IFL) has largely been driven by the design of specialized deep architectures to capture manipulation artifacts. CNN-based approaches Huang et al. (2025); Cui et al. (2025); Zhu et al. (2024); Dong et al. (2022) typically focus on modeling local anomalies such as texture inconsistencies, boundary discontinuities, and noise residuals. For instance, MVSS-Net Dong et al. (2022) integrates an edge-supervised branch with a noise-sensitive branch to detect forgery edges and residual noise, showing the benefit of multi-branch forensic cues. While effective, these CNN-based methods are inherently limited by local receptive fields and often fail to capture long-range structural dependencies. To address this, Transformer-based solutions Zeng et al. (2024); Wang et al. (2022); Lou et al. (2024); Ma et al. (2023); Kong et al. (2025); Hao et al. (2021) have recently emerged, leveraging global self-attention to model long-range pixel relationships. For example, IML-ViT Ma et al. (2023) fully fine-tunes a pre-trained Vision Transformer for IFL, demonstrating the promise of large-scale pre-training in forensic scenarios. Nevertheless, such approaches frequently require heavy backbone modification or full fine-tuning, which not only incurs high computational costs but also risks catastrophic forgetting of upstream priors. Moreover, their reliance on curated or synthetic datasets limits robustness when facing diverse, real-world forgeries. In contrast, our work proposes a parameter-efficient paradigm that resolves the *representation conflict* between content-oriented VFMs and forgery-sensitive forensic tasks. Instead of redesigning or fully tuning the backbone, we introduce In-Context Alignment (ICA), which injects dual prompts into frozen VFMs, to preserve semantic priors and enhance generalization.

### 2.2 VISUAL PROMPT LEARNING

Prompt learning Zhang et al. (2025); Shao et al. (2025) has emerged as an efficient strategy to adapt large pre-trained models by training only a small set of learnable parameters. VPT Jia et al. (2022) prepends task-specific prompt tokens to the input sequence of a frozen Transformer, while EVP Liu et al. (2023a) incorporates frequency-domain prompts into intermediate layers to enhance foreground segmentation. In the forensic domain, CLIP-IFDL Li et al. (2024a) extends CLIP with a noise-aware adapter to improve manipulation detection. These approaches highlight the potential of prompts as lightweight adapters, yet most methods focus on either semantic or noise cues in isolation, leaving the diversity of forgery traces underexplored. Our approach differs in two key aspects. First, we explicitly decompose forgery cues into *physical* (e.g., noise, frequency) and *semantic* (e.g., contextual plausibility) components. Second, we introduce a dual prompting mechanism, the Physical-Aware Prompter (PAP) and the Semantic-Aware Prompter (SAP), that jointly address both perspectives. This design offers a more comprehensive and robust approach to forgery localization, while maintaining the parameter efficiency and generalization benefits of prompt-based learning.

## 3 METHODOLOGY

### 3.1 MOTIVATION AND PROBLEM FORMULATION

The unparalleled efficacy of VFMs stems from their large-scale pre-training, which produces content-prioritized representations highly effective for semantic tasks. However, this optimization induces a systematic bias that is fundamentally misaligned with forgery-sensitive tasks such as image forgery localization (IFL). This misalignment is termed the *representation conflict*, which manifests as two complementary biases: **Physical Suppression Bias**, where low-level forensic cues (e.g., sensor noise, compression artifacts, frequency inconsistencies) are marginalized as nuisances during pre-training, creating a *modality gap* between RGB-focused features and forensically relevant

domains, and Semantic Overconfidence Bias, where pretrained models enforce global coherence, often *explaining away* localized inconsistencies as natural variations, leading to a *semantic gap* that conceals subtle manipulation artifacts. Formally, let $\mathcal{R}_{pre}$ denote the pretrained VFM representation space and $\mathcal{R}_{forg}$ the desired forgery-sensitive space. The conflict can be expressed as the divergence:

$$\mathcal{C} = \mathbb{E}_{x \sim \mathcal{X}} \left[ \| f_{pre}(x) - f_{forg}(x) \|_2 \right],\tag{1}$$

where $f_{pre}$ and $f_{forg}$ are the pretrained and oracle forgery-sensitive mappings, respectively, and $\mathcal{X}$ is the data distribution. This $\mathcal{C}$ can be decomposed as $\mathcal{C} \approx \mathcal{C}_{physical} + \mathcal{C}_{semantic}$, where $\mathcal{C}_{physical}$ quantifies the modality gap and $\mathcal{C}_{semantic}$ the semantic gap. Conventional full fine-tuning minimizes $\mathcal{C}$ by updating all parameters, but this approach is computationally expensive, prone to catastrophic forgetting (with generalization bounds $O(\sqrt{|\theta|/n})$, where $|\theta|$ is the number of parameters and $n$ the sample size), and sample-inefficient. The core problem is the inefficiency in resolving $\mathcal{C}$ without disrupting valuable priors. Our objective is to design a parameter-efficient strategy that learns minimal, input-aware perturbations, implemented as prompts, to align $\mathcal{R}_{pre}$ with $\mathcal{R}_{forg}$ while preserving priors. This reformulates IFL as a *visual in-context alignment problem*, where prompts serve as lightweight interventions to minimize:

$$\arg\min_{\phi} \mathbb{E}_x \left[ \| f_{pre}(x \oplus g(\phi; x)) - f_{forg}(x) \|_2 \right] + \lambda \Omega(\phi),\tag{2}$$

with generalization bounds $O(\sqrt{|\phi|/n})$, where $|\phi| \ll |\theta|$, ensuring data efficiency and robustness. $\Omega(\phi)$ regularizes prompt complexity to prevent overfitting and ensure minimal changes.

Decomposing the objective naturally leads to dual-stream solutions: specialized prompts for each sub-conflict, integrated to handle diverse forgeries. Consequently, we instantiate this formulation through the In-Context Alignment framework, where the prompt function $g(\phi; x)$ is realized via two complementary prompters with a small parameter set $\phi$ that enables parameter-efficient adaptation.

## 3.2 OVERVIEW

Our approach, termed In-Context Alignment (ICA), achieves this goal through dual-awareness prompting mechanisms. As illustrated in Figure 2, ICA incorporates two complementary modules to resolve the decomposed conflict: the Physical-Aware Prompter (PAP), which bridges the modality gap by enhancing sensitivity to low-level forensic cues via a Mixture-of-Experts design for adaptive fusion, minimizing $\mathcal{C}_{physical}$, and the Semantic-Aware Prompter (SAP), which acts as a contextual probe to expose semantic implausibilies, minimizing $\mathcal{C}_{semantic}$. By integrating these modules into a frozen VFM backbone, ICA preserves pretrained knowledge while aligning forensic and semantic evidence for robust forgery localization with minimal parameter cost. This paradigm not only mitigates the representation conflict but also extends to broader forgery detection tasks.

## 3.3 PRELIMINARIES

Vision Foundation Models (VFMs), such as Vision Transformer (ViT) Dosovitskiy et al. (2021); Li et al. (2022); Oquab et al. (2023) and SegFormer Xie et al. (2021), are pre-trained on large-scale vision tasks and consist of an encoder backbone for feature extraction and a decoder for task-specific outputs. Given a manipulated image $I \in \mathbb{R}^{H \times W \times 3}$, the encoder splits it into $M$ patches $\{I^i\}_{i=1}^M$, projecting them into $d$-dimensional tokens $\{t_0^i \in \mathbb{R}^d\}_{i=1}^M$. These tokens form the initial embedding $x_0 \in \mathbb{R}^{M \times d}$. The embedding is processed through $N$ transformer layers via $x_i = L_i(x_{i-1})$. In ICA, our objective is to adapts VFMs for forgery localization by constructing a model $\hat{T} = \{T, \mathcal{P}\}$, where $T$ is the frozen VFM backbone (with parameters $\theta$) and $\mathcal{P}$ is a small set of trainable prompts parameterized by $\phi$ ($|\phi| \ll |\theta|$). By injecting $\mathcal{P}$ into the input sequence using the hybrid operator $\oplus$, ICA refines token embeddings to detect manipulated regions while retaining pretrained priors.

## 3.4 IN-CONTEXT ALIGNMENT FOR IFL

In-Context Alignment (ICA) addresses representation conflicts in VFMs for image forgery localization (IFL) through a dual-stream prompting framework, as shown in Figure 2. By decomposing the conflict space $\mathcal{C}$ into physical ($\mathcal{C}_{physical}$) and semantic ($\mathcal{C}_{semantic}$) components, ICA integrates two complementary modules: the Physical-Aware Prompter (PAP) and the Semantic-Aware Prompter

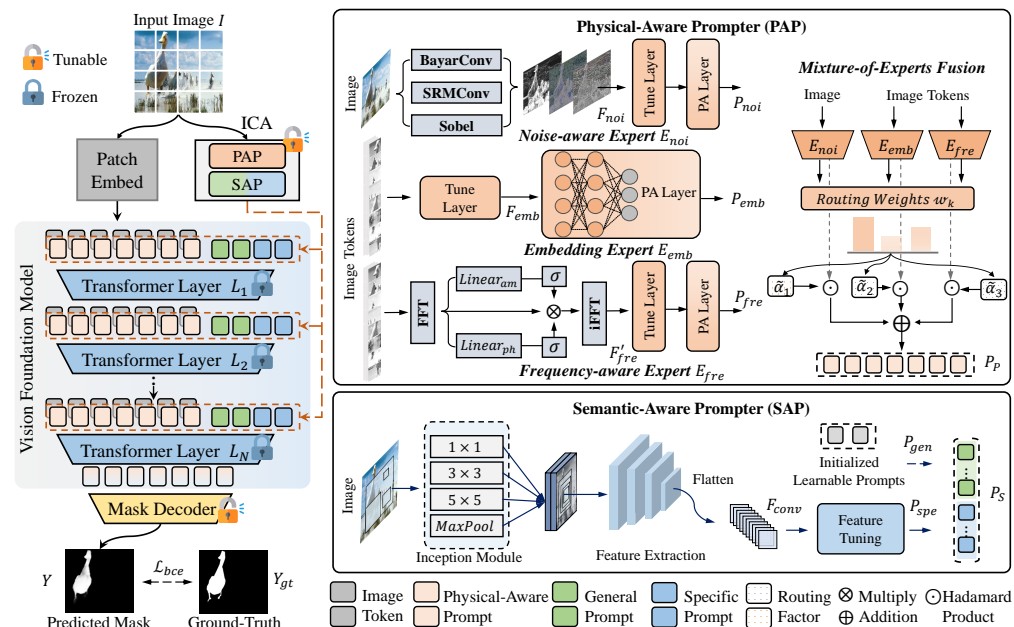

Figure 2: In-Context Alignment (ICA) pipeline for image forgery localization. ICA refines a frozen Vision Foundation Model by injecting lightweight Physical-Aware Prompter (PAP) and Semantic-Aware Prompter (SAP) prompts into the input sequence of transformer layers. PAP amplifies low-level forensic cues (e.g., noise inconsistencies) using a Mixture-of-Experts fusion mechanism, while SAP captures high-level semantic inconsistencies through contextual guidance.

(SAP). These prompters collectively form the lightweight prompt function $g(\phi; x)$, where $\phi$ denotes minimal trainable parameters ($|\phi| \ll |\theta|$, with $\theta$ representing the VFM's parameters), ensuring parameter efficiency while preserving priors. The prompts refine token embeddings in the frozen VFM to bridge physical and semantic gaps. Specifically, the PAP enhances low-level forensic cues (e.g., noise inconsistencies) suppressed by VFMs' content-prioritized representations. It generates physical prompts $\mathcal{P}_P = \{P_P^{i-1}\}_{i=1}^N$, where $P_P^{i-1}$ is the prompt for the input to the $i$-th encoder layer, and $N$ is the number of transformer layers. We then perform additive injection where these prompts are added to the input sequence, shifting representations toward a forensic-sensitive subspace and minimizing $\mathcal{C}_{\text{physical}}$ by amplifying forgery-sensitive signals. Conversely, the SAP addresses high-level semantic implausibilities by providing contextual guidance. It produces semantic prompts $\mathcal{P}_S = \{P_S^{i-1}\}_{i=1}^N$, which are concatenated to the input sequence, enabling self-attention to capture semantic inconsistencies and minimize $\mathcal{C}_{\text{semantic}}$. To instantiate the abstract prompt injection $x \oplus g(\phi; x)$ from the optimization objective, the layer-wise input is defined as $x_i = [x_{i-1} + P_P^{i-1}, P_S^{i-1}] = x_{i-1} \oplus g(\phi; x)$, where $\phi = \{\phi_P, \phi_S\}$ denotes the trainable parameters for prompt generation. The hybrid operator $\oplus$ combines addition ($+$) for physical modulation (e.g., adjusting features) and concatenation ($[\cdot, \cdot]$) for semantic extension (e.g., incorporating contextual information). The enhanced input $x_i$ is processed by the $i$-th transformer layer to yield $x_{i+1} = L_i(x_i)$. After processing through all $N$ layers, the output embeddings are fed into a lightweight mask decoder to predict the forgery mask $Y$. By integrating PAP and SAP, ICA aligns low-level forensic cues with high-level semantic reasoning, resolving representation conflicts with minimal parameter overhead, as the small $|\phi|$ ensures efficiency while preserving VFM prior knowledge.

### 3.5 PHYSICAL-AWARE PROMPTER (PAP)

The Physical-Aware Prompter (PAP) mitigates the suppression of low-level forensic cues (e.g., noise or frequency artifacts) in VFMs, which contributes to the physical conflict space $\mathcal{C}_{\text{physical}}$. As shown in the PAP component of Figure 2, PAP generates input-adaptive prompts to amplify these cues, minimizing the modality gap. It integrates three specialized experts: Noise-Aware Expert, Embedding Expert, and Frequency-Aware Expert, whose outputs are combined via a Mixture-of-Experts (MoE) mechanism to ensure robustness across diverse forgery patterns.

**Noise-Aware Expert.** To extract noise-related forensic cues, we first apply specialized filters to the input image $I \in \mathbb{R}^{H \times W \times 3}$: BayarConv Bayar & Stamm (2018) adaptively learns manipulation traces, SRMConv Fridrich & Kodovsky (2012) detects structural edges and textures resilient to compression, and the Sobel filter captures strong boundaries. These filters yield noise-aware features:

$$F_{noi} = Concat([F_{bayar}(I), F_{srm}(I), F_{sobel}(I)]), \tag{3}$$

Next, we adopt a patch embedding strategy Xie et al. (2021) that leverages a convolutional layer to enable robust token extraction. $F_{noi}$ is partitioned into $M$ patches matching the VFM backbone's patch size and projected into a $c$-dimensional space via a tunable linear layer (*Tune Layer*), yielding $F'_{noi} \in \mathbb{R}^{M \times c}$, where $c = d/r$ and $r$ is a scale factor controlling parameter efficiency. To ensure effective adaptation across all layers, we introduce a Prompt Adjustment (PA) Layer consisting of two lightweight MLPs with GELU activation. The first MLP layer learns distinct prompts for each PA layer, while the second layer, $MLP_s$, is a shared up-projection layer across all PA layers to match the backbone feature dimensions. Each PA layer uses $F'_{noi}$ to derive the noise-aware prompt:

$$P_{noi} = MLP_s(GELU(MLP(F'_{noi}))) \in \mathbb{R}^{M \times d}, \tag{4}$$

**Embedding Expert.** To enhance the frozen patch embeddings of the VFM, we adapt the initial embedding $x_0 \in \mathbb{R}^{M \times d}$ derived from $I$. A *Tune Layer* projects $x_0$ into a lower-dimensional space, producing $F_{emb} \in \mathbb{R}^{M \times c}$, where $c = d/r$. This feature is then processed by a PA Layer identical to that in the Noise-Aware Expert, generating the tuned and adapted embedding prompt as follows:

$$P_{emb} = MLP_s(GELU(MLP(F_{emb}))) \in \mathbb{R}^{M \times d}, \tag{5}$$

This prompt refines the VFM's content-prioritized embeddings to capture subtle manipulation traces, enhancing forgery sensitivity without altering the frozen backbone.

**Frequency-Aware Expert.** While many frequency-based methods rely on direct frequency extraction, which often leads to overfitting to specific training data, our approach introduces a novel frequency-space learning method that enhances forgery traces through selective modulation of frequency components. First, the image tokens $x_0 \in \mathbb{R}^{M \times d}$ are derived and transformed into the frequency domain using the Fast Fourier Transform ($\mathcal{FFT}$) as:

$$F_{fre} = F_{am} + F_{ph} \cdot j = \mathcal{FFT}(x_0), \tag{6}$$

where $F_{am}$ and $F_{ph}$ are the amplitude and phase spectra, respectively and $j$ is the imaginary unit. Linear layers recalibrate each spectrum to amplify forgery-relevant frequencies as follows:

$$F'_{am} = F_{am} \cdot \sigma(Linear_{am}(F_{am})), \quad F'_{ph} = F_{ph} \cdot \sigma(Linear_{ph}(F_{ph})), \tag{7}$$

where $\sigma(\cdot)$ is the Sigmoid function. The recalibrated spectra are converted back to the feature space via the inverse FFT ($\mathcal{IFFT}$), producing enhanced embeddings $F'_{fre}$ as follows:

$$F'_{fre} = \mathcal{IFFT}(F'_{am} + F'_{ph} \cdot j), \tag{8}$$

The enhanced $F'_{fre}$ is subsequently projected into a $c$-dimensional feature space through a *Tune Layer*, resulting in $F''_{fre} \in \mathbb{R}^{M \times c}$. Similar to the PA layer used in the Noise-Aware Expert, $F''_{fre}$ is processed through a PA layer to generate the frequency-aware prompt defined as:

$$P_{fre} = MLP_s(GELU(MLP(F''_{fre}))) \in \mathbb{R}^{M \times d}, \tag{9}$$

**Mixture-of-Experts Fusion.** To adaptively integrate the noise-aware ($P_{noi}$), embedding ($P_{emb}$), and frequency-aware ($P_{fre}$) prompts, PAP employs a Mixture-of-Experts (MoE) design with a parameter-free cosine router that performs lightweight, input-aware reweighting. Given the layer input $x_{i-1}$ and expert prompts $\{P_k\}_{k=1}^{K}$ ($K = 3$), we compute cosine similarities as:

$$s_k(x_{i-1}) = \left\langle \frac{x_{i-1}}{\|x_{i-1}\|}, \frac{P_k}{\|P_k\|} \right\rangle, \qquad w_k(x_{i-1}) = \mathrm{Softmax}\left( \frac{s_k(x_{i-1})}{\tau} \right), \tag{10}$$

where $\tau$ is a temperature. We then form hybrid expert weights by a convex interpolation between learnable coefficients $\alpha_k$ and the cosine router weights, controlled by a factor $\beta$ (set to 0.5) as:

$$\tilde{\alpha}_k(x_{i-1}) = (1 - \beta)\,\alpha_k + \beta\,w_k(x_{i-1}), \tag{11}$$

Finally, the fused physical prompt can be obtained as:

$$P_P^{i-1} = \sum_{k=1}^{K} \tilde{\alpha}_k(x_{i-1})\,P_k, \quad P_k \in \{P_{noi}, P_{emb}, P_{fre}\}. \tag{12}$$

Overall, the final $P_P$ is produced by an adaptive fusion over multiple experts, enabling effective integration of multi-view forensic cues and yielding robust generalization to diverse forgery types.

### 3.6 SEMANTIC-AWARE PROMPTER (SAP)

The Semantic-Aware Prompter (SAP) addresses the semantic overconfidence bias in Vision Foundation Models (VFMs), where pretrained representations enforce global coherence, often masking localized semantic implausibilities (e.g., illogical object interactions). SAP minimizes $\mathcal{C}_{semantic}$ by generating contextual prompts that guide the VFM to probe high-level inconsistencies, as shown in the SAP component of Figure 2. SAP produces two types of prompts: instance-specific prompts ($P_{spe}$) to capture localized forgery cues and general prompts ($P_{gen}$) to encode universal forgery characteristics, forming the complete semantic prompt set $\mathcal{P}_S = \{P_S^{i-1}\}_{i=1}^N$.

**Instance-Specific Prompt Generation.**   To capture localized forgery cues, SAP employs a two-step process: Feature Extraction and Tuning. In Feature Extraction, the input image $I \in \mathbb{R}^{H \times W \times 3}$ is processed by a multi-branch Inception Module with $1 \times 1$, $3 \times 3$, $5 \times 5$ convolutions and Max-Pooling, to extract multi-scale local features. These features are fed into a shallow ConvNet with sequential $3 \times 3$ stride-2 convolutional layers, progressively doubling the channel dimensions while reducing spatial resolution. The output is adaptively pooled to a fixed resolution $h \times w$, yielding:

$$F_{conv} = Pool(ConvNet(Inception(I))) \in \mathbb{R}^{h \times w \times c_0}, \tag{13}$$

where $c_0$ is the output channel dimension. In Feature Tuning, $F_{conv}$ is flattened and projected into a $d$-dimensional sequence via a tunable linear layer, aligning with the VFM's token dimensionality:

$$P_{spe} = FeatureTuning(F_{conv}) \in \mathbb{R}^{n \times d}, \tag{14}$$

where $n = h \times w$ is the number of spatial tokens. $P_{spe}$ provides instance-specific forensic cues, enabling the VFM to detect localized semantic inconsistencies.

**General Prompt Generation.**   To enhance generalization to unseen forgery types, SAP introduces $m$ learnable tokens as the general prompt $P_{gen} \in \mathbb{R}^{m \times d}$, inspired by Visual Prompt Tuning Jia et al. (2022). These tokens are randomly initialized and shared across all training images, capturing universal forgery characteristics. During training, $P_{gen}$ interacts with image tokens in each transformer layer via self-attention, learning forgery patterns in a data-driven manner. The complete semantic prompt is formed by concatenating the instance-specific and general prompts:

$$P_S^{i-1} = [P_{spe}, P_{gen}] \in \mathbb{R}^{(n+m) \times d}, \tag{15}$$

where the prompts are shared across layers for efficiency. SAP enhances the VFM's ability to detect semantic implausibilities, minimizing $\mathcal{C}_{semantic}$ while maintaining parameter efficiency.

### 3.7 COMPLEXITY AND OPTIMIZATION

The ICA framework introduces minimal additional parameters through $\mathcal{P}_P$ (PAP) and $\mathcal{P}_S$ (SAP), ensuring parameter efficiency. For instance, with a SegFormer-B4 backbone Xie et al. (2021) (64M parameters), PAP contributes $\approx 0.35$M parameters, and SAP contributes $\approx 1.67$M parameters, amounting to only 0.55% and 2.61% of the total model parameters, respectively. Only the prompts and localization decoder are trainable, while the VFM backbone remains frozen, significantly reducing storage and computational costs. During training, we optimize the prompt parameters and decoder using binary cross-entropy (BCE) loss, where $Y$ is the predicted mask and $Y_{gt}$ is the GT:

$$\arg \min_{\mathcal{P}_P, \mathcal{P}_S, \theta_{decoder}} \mathcal{L}_{bce}(Y, Y_{gt}). \tag{16}$$

To ensure balanced utilization of the three experts in PAP's Mixture-of-Experts (MoE) fusion, we introduce a load-balancing loss to encourage a uniform routing distribution:

$$\bar{\mathbf{w}} = \frac{1}{B} \sum_{b=1}^{B} w_b, \quad \mathcal{L}_{bal} = \left\| \bar{\mathbf{w}} - \frac{1}{K} \mathbf{1} \right\|_2^2, \tag{17}$$

where $w_b \in \mathbb{R}^K$ ($K = 3$) is the cosine router's weight vector for the $b$-th sample, $B$ is the batch size, and $\frac{1}{K} \mathbf{1}$ is the uniform distribution over experts. The total loss combines both terms:

$$\mathcal{L} = \mathcal{L}_{bce} + \lambda \mathcal{L}_{bal}. \tag{18}$$

with $\lambda = 0.1$ balancing the contributions. This optimization strategy minimizes the objective introduced in problem formulation by learning the parameters $\phi$ of the prompt function $g(\phi; x)$, which is composed of $\mathcal{P}_P$ and $\mathcal{P}_S$, thereby aligning the pretrained representation space $\mathcal{R}_{pre}$ to the forgery-sensitive space $\mathcal{R}_{forg}$, achieving SoTA performance with minimal data and computational demands.

Table 1: Comparison of SoTA methods on cross-dataset pixel-level localization. Reported metrics are F1 scores(%)(↑), with best results in bold and quoted results marked by '*'.

| Methods | NIST16 | Columbia | CASIAv1+ | IMD | COVER | DSO-1 | DEF-12K | In-Wild | Korus | AVG |
|---|---|---|---|---|---|---|---|---|---|---|
| *Full model training or tuning methods.* | | | | | | | | | | |
| FCN Long et al. (2015) | 16.70 | 22.30 | 44.10 | 21.00 | 19.90 | 6.80 | 13.00 | 19.20 | 12.20 | 19.50 |
| U-Net Ronneberger et al. (2015) | 17.30 | 15.20 | 24.90 | 14.80 | 10.70 | 12.40 | 4.50 | 17.50 | 11.70 | 14.30 |
| DeepLabv3 Chen et al. (2017) | 23.70 | 44.20 | 42.90 | 21.60 | 15.10 | 16.40 | 6.80 | 22.00 | 12.00 | 22.70 |
| MFCN* Salloum et al. (2018) | 24.30 | 18.40 | 34.60 | 17.00 | 14.80 | 15.00 | 6.70 | 16.10 | 11.80 | 17.60 |
| RRU-Net Bi et al. (2019) | 20.00 | 26.40 | 29.10 | 15.90 | 7.80 | 8.40 | 3.30 | 17.80 | 9.70 | 15.40 |
| ManTra-Net Wu et al. (2019) | 15.80 | 45.20 | 18.70 | 16.40 | 23.60 | 25.50 | 6.70 | 31.40 | 11.00 | 21.60 |
| HPFCN* Li & Huang (2019) | 17.20 | 11.50 | 17.30 | 11.10 | 10.40 | 8.20 | 3.80 | 12.50 | 9.70 | 11.30 |
| H-LSTM* Bappy et al. (2019) | 35.70 | 14.90 | 15.60 | 20.20 | 16.30 | 14.20 | 5.90 | 17.30 | 14.30 | 17.20 |
| SPAN Hu et al. (2020) | 21.10 | 50.30 | 14.30 | 14.50 | 14.40 | 8.20 | 3.60 | 19.60 | 8.60 | 17.20 |
| ViT-B Dosovitskiy et al. (2021) | 25.40 | 21.70 | 28.20 | 15.40 | 14.20 | 16.90 | 6.20 | 20.80 | 17.60 | 18.50 |
| Swin-ViT Liu et al. (2021) | 22.00 | 36.50 | 39.00 | 30.00 | 16.80 | 18.30 | 15.70 | 26.50 | 13.40 | 24.20 |
| SegFormer-B4 Xie et al. (2021) | 21.80 | 31.60 | 41.60 | 21.60 | 10.00 | 15.10 | 6.40 | 21.70 | 10.80 | 20.10 |
| PSCC-Net Liu et al. (2022) | 17.30 | 50.30 | 33.50 | 19.70 | 22.00 | 29.50 | 7.20 | 30.30 | 11.40 | 24.60 |
| MVSS-Net++ Dong et al. (2022) | 30.40 | 66.00 | 51.30 | 27.00 | 48.20 | 27.10 | 9.50 | 29.50 | 10.20 | 33.20 |
| CAT-Net Kwon et al. (2022) | 10.20 | 20.60 | 23.70 | 25.70 | 21.00 | 17.50 | 20.60 | 21.70 | 8.50 | 18.80 |
| TruFor* Ma et al. (2023) | 26.80 | 82.90 | 53.20 | 35.90 | 28.00 | 21.30 | 14.80 | 36.10 | 12.20 | 34.60 |
| MPC* Lou et al. (2024) | 29.10 | 67.60 | 44.80 | 48.50 | 41.00 | 36.90 | 22.00 | 43.30 | 25.10 | 39.80 |
| IML-ViT Ma et al. (2023) | 33.10 | 78.00 | 72.10 | 32.70 | 43.50 | 7.70 | 21.60 | 16.70 | 4.70 | 36.30 |
| PIM* Kong et al. (2025) | 28.00 | 68.00 | 56.60 | 41.90 | 25.10 | 25.30 | 16.70 | 41.80 | 23.40 | 36.30 |
| *Parameter-efficient tuning methods.* | | | | | | | | | | |
| VPT-Deep Jia et al. (2022) | 25.50 | 47.30 | 50.40 | 31.20 | 8.80 | 1.90 | 17.10 | 19.30 | 11.00 | 23.60 |
| AdaptFormer Chen et al. (2022) | 26.90 | 78.60 | 53.30 | 36.40 | 22.40 | 5.80 | 19.10 | 27.30 | 12.10 | 31.30 |
| EVPv1 Liu et al. (2023a) | 29.40 | 63.80 | 54.30 | 35.00 | 18.20 | 4.60 | 16.80 | 29.10 | 13.30 | 29.40 |
| EVPv2 Liu et al. (2023b) | 30.50 | 61.00 | 55.70 | 38.00 | 22.80 | 15.10 | 17.10 | 29.60 | 16.20 | 31.80 |
| ICA (Ours) | **34.90**(+4.4) | **81.50**(+2.9) | **77.60**(+21.9) | **41.80**(+3.8) | **33.40**(+10.6) | **19.20**(+4.1) | **22.50**(+3.4) | **33.00**(+3.4) | **19.40**(+3.2) | **40.30**(+8.5) |

## 4 EXPERIMENTS

### 4.1 EXPERIMENTAL SETTINGS

**Dataset.** To ensure comprehensive and fair comparisons with existing algorithms, we conduct experiments utilizing multiple scales of benchmark datasets with various tampering types, encompassing both homologous datasets and cross-dataset evaluations. The two types of experiments are described as follows: In cross-dataset experiments, following previous studies Dong et al. (2022); Kong et al. (2025); Lou et al. (2024), all models are trained exclusively on the CASIAv2 Dong et al. (2013) dataset and subsequently test on the 9 additional datasets: CASIAv1+ Dong et al. (2013), NIST16 Guan et al. (2019), Columbia Hsu & Chang (2006), IMD Novozamsky et al. (2020), COVER Wen et al. (2016), DSO-1 Carvalho et al. (2015), DEF-12K Mahfoudi et al. (2019), In-Wild Huh et al. (2018), and Korus Korus & Huang (2016). In homologous dataset experiments, following Wu et al. (2019); Zhou et al. (2023), our model uses only four benchmark datasets for training and evaluation as detailed in the *Supplementary Material* A.

**Implementation Details.** The ICA framework is implemented on PyTorch, utilizing the AdamW optimizer Loshchilov & Hutter (2019) with a learning rate of $5 \times 10^{-4}$. Training is conducted with a batch size of 8 and an input crop size of $512 \times 512$, using a cosine decay schedule for the learning rate. The PAP produces $P_P$ tokens aligned with the number of image tokens, while the SAP generates $P_{spe}$ and $P_{gen}$ tokens, empirically set to 64 and 10, respectively. A scale factor of $r = 4$ controls prompt dimensionality to enhance parameter efficiency. Parameter-efficient tuning methods are built upon the SegFormer-B4 backbone Xie et al. (2021) with a lightweight MLP-based mask decoder. Additional experimental results and analysis are provided in the *Supplementary Material* A.

### 4.2 COMPARISON WITH STATE-OF-THE-ART METHODS

**Cross-dataset Experiments.** We conduct a comprehensive cross-dataset evaluation of ICA against 19 existing full fine-tuning forgery localization methods and segmentation algorithms. In addition, we also consider comparisons with parameter-efficient tuning-based methods. Table 1 shows the pixel-level F1 performance of these methods. Despite the 9 testing datasets exhibiting diverse distributions, ICA achieves the highest average F1 score, which outperforms previous methods and demonstrates its strong generalization capability across diverse datasets. While some parameter-efficient methods show reasonable results, ICA consistently performs better across most datasets, These results highlight ICA's high localization precision and robustness in real-world applications.

**Homologous-dataset Experiments.** Following Zhou et al. (2023), we validate the efficacy of our approach across various datasets, including traditional approaches and deep learning-based models, many of which utilize pre-training on additional synthetic datasets. To ensure consistent comparison

Table 2: Homologous-dataset evaluation with pixel-level AUC scores. AT: Additional Training Data, CAS.: CASIAv1+, COV.: COVER, NIS.: NIST, Col.: Columbia.

| Methods | AT | COV. | CAS. | NIS. | Col. | IMD | AVG |
|---|---|---|---|---|---|---|---|
| ELA Abd Warif et al. (2015) | × | 58.3 | 61.3 | 42.9 | 58.1 | - | - |
| NOI Mahdian & Saic (2009) | × | 58.7 | 61.2 | 48.7 | 54.6 | - | - |
| CFA Ferrara et al. (2012) | × | 48.5 | 52.2 | 50.1 | 72.0 | - | - |
| J-LSTM Bappy et al. (2017) | ✓ | 61.4 | - | 76.4 | - | - | - |
| ManTra-Net Wu et al. (2019) | 64K | 81.9 | 81.7 | 79.5 | 82.4 | - | - |
| RGB-N Zhou et al. (2018) | 42K | 81.7 | 79.5 | 93.7 | 85.8 | - | - |
| SPAN Hu et al. (2020) | 96K | 93.7 | 83.8 | 96.1 | - | 75.0 | - |
| ObjectFormer Wang et al. (2022) | 62K | 95.7 | 88.2 | 99.6 | - | - | - |
| IF-OSN Wu et al. (2022) | ✓ | 88.3 | 83.3 | 76.4 | - | - | - |
| PSCC-Net Liu et al. (2022) | 100K | 94.1 | 87.5 | 99.1 | - | 80.6 | 90.3 |
| MVSS-Net Chen et al. (2021) | 60K | 82.4 | 75.3 | 73.7 | 72.6 | 53.8 | 71.6 |
| MVSS-Net++ Dong et al. (2022) | 60K | 52.5 | 77.1 | 71.5 | 56.3 | 88.6 | 69.2 |
| PCL Zeng et al. (2023) | 100K | 91.7 | 75.1 | 94.6 | 76.1 | 82.3 | 84.0 |
| TruFor Guillaro et al. (2023) | ✓ | 92.5 | 95.7 | 87.7 | 94.7 | - | - |
| HiFiNet Guo et al. (2023) | 100K | 93.2 | 85.8 | 87.0 | 98.3 | 82.9 | 89.4 |
| NGNet Zhu et al. (2024) | 60K | 94.1 | 87.2 | 90.0 | 98.5 | 85.2 | 91.0 |
| NCL Zhou et al. (2023) | × | 92.8 | 86.4 | 91.2 | 94.3 | 88.9 | 90.7 |
| EVPv1 Liu et al. (2023a) | × | 89.0 | 89.1 | 88.5 | 96.4 | 86.3 | 89.8 |
| Adaptformer Chen et al. (2022) | × | 84.2 | 89.9 | 87.1 | 98.1 | 83.7 | 88.6 |
| ICA (Ours) | × | **98.5** | **96.9** | 96.0 | **99.9** | **90.2** | **96.3** |

Table 3: Robustness analysis under common post-processing distortions. We report pixel-level AUC scores (higher is better), including (i) *Resize*, scale factor ×), (ii) Gaussian blur with kernel size $k$, (iii) additive Gaussian noise with standard deviation $\sigma$, and (iv) JPEG compression with quality factor $q$.

| Distortion | SPAN | ObjectFormer | NCL | Ours |
|---|---|---|---|---|
| w/o distortion | 0.836 | 0.872 | 0.912 | **0.960** |
| Resize(0.78×) | 0.832 | 0.872 | 0.856 | **0.936** |
| Resize(0.25×) | 0.803 | 0.863 | 0.831 | **0.889** |
| GaussianBlur($k = 3$) | 0.831 | 0.860 | 0.840 | **0.950** |
| GaussianBlur($k = 15$) | 0.792 | 0.803 | 0.806 | **0.874** |
| GaussianNoise($\sigma = 3$) | 0.752 | 0.796 | 0.795 | **0.818** |
| GaussianNoise($\sigma = 15$) | 0.673 | 0.782 | 0.714 | **0.737** |
| JPEGCompress($q = 100$) | 0.836 | 0.864 | 0.843 | **0.960** |
| JPEGCompress($q = 50$) | 0.807 | 0.862 | 0.819 | **0.953** |

Table 4: Ablation study of In-Context Alignment (ICA) on pixel-level F1 scores (%).

| Settings | Trainable Param. (M) | Pixel-level F1 (%) | | | | |
|---|---|---|---|---|---|---|
| | | NIS | CAS | COV | Col | AVG |
| Full-tuning | 64.00 | 24.6 | 56.6 | 19.4 | 34.6 | 33.8 |
| OnlyDecoder | 3.15 | 19.3 | 26.3 | 11.3 | 19.1 | 19.0 |
| Decoder + $\mathcal{P}_{emb}$ | 3.23 (+0.08) | 25.3 | 54.9 | 16.9 | 22.6 | 29.9 |
| + $\mathcal{P}_{noi}$ | 3.33 (+0.10) | 25.9 | 52.5 | 19.9 | 31.3 | 32.4 |
| + $\mathcal{P}_{freq}$ (PAP) | 3.50 (+0.17) | 24.8 | 63.3 | 26.4 | 36.8 | 37.8 |
| + $\mathcal{P}_{spe}$ | 5.06 (+1.56) | 28.7 | 63.6 | 28.8 | 43.0 | 41.0 |
| + $\mathcal{P}_{gen}$ (PAP + SAP) | 5.17 (+0.11) | 31.2 | 67.6 | 23.7 | 45.3 | 41.9 |
| w/o MoE (static weights) | 5.17 (+0.00) | 29.8 | 65.5 | 22.9 | 43.9 | 40.5 |
| w/o $\mathcal{L}_{bal}$ | 5.17 (+0.00) | 30.1 | 64.8 | 23.2 | 44.2 | 40.6 |
| w/o shared MLP | 5.87 (+0.70) | 31.4 | 66.3 | 26.8 | 47.9 | 43.1 |

Figure 3: Impact of scale factor $r$ on performance on CASIAv1+ dataset.

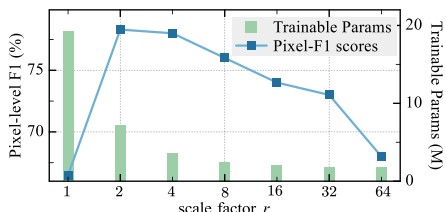

with most methods, we report AUC as the global metric, as it is less sensitive to pixel imbalance. As shown in Table 2, traditional methods yield low AUC scores, while deep models with pre-training improve performance. Certain designs, like NCL achieve strong results without extra data, showing the importance of architecture. Our ICA attains the highest average score even without pre-training.

## 4.3 ROBUSTNESS EVALUATION

We apply different image distortion methods on raw images from the NIST16 dataset and report AUC scores to evaluate the robustness of our ICA. We include four types of distortions: 1) resizing images to different scales, 2) applying Gaussian blur with a kernel size $k$, 3) adding Gaussian noise with a deviation $\sigma$, and 4) applying JPEG compression with a quality factor $q$. Table 3 shows that our method exhibits robustness to various distortion operations, outperforming others.

## 4.4 ABLATION STUDIES

We conduct ablation studies to assess the impact of each key component and its trainable parameters, with results detailed in Table 4. It shows that full fine-tuning is inefficient, while adding tailored prompts progressively boosts performance with minimal parameter overhead. Physical- and semantic-aware prompts deliver the largest gains, and further analysis confirms that MoE routing and balanced loss are crucial for stable improvements. Besides, we analyze the impact of the scale factor $r$, which controls tunable parameters. As shown in Figure 3, increasing $r$ leads to fewer trainable parameters. At $r = 4$, our model achieves an optimal balance between accuracy and efficiency.

## 5 CONCLUSION

In this paper, we address the representation conflict in VFMs for image forgery localization, where physical and semantic cues are suppressed, limiting generalization. We propose In-Context Alignment, a parameter-efficient framework that resolves this conflict through dual-stream prompting. The Physical-Aware Prompter adaptively amplifies low-level forensic cues via MoE fusion, while the Semantic-Aware Prompter audits high-level inconsistencies using specific and general prompts. With a small fraction of parameters, ICA achieves robust generalization across diverse forgery types.

**Reproducibility Statement.** We have made efforts to ensure the reproducibility of our work through several measures. All datasets used in this paper are publicly available. We list dataset names and citations in Section 4 and the Appendix A. No private or proprietary data is used. We report crucial hyperparameters, model sizes, pre-processing, and training schedules in Section 4 and the Appendix A. We fix random seeds for all frameworks where applicable and disable nondeterministic kernels when possible. These resources and descriptions are intended to allow independent researchers to replicate our results with minimal effort.

**Ethics Statement.** This work uses only publicly available datasets under their respective licenses and does not involve the collection of new human subjects data. To the best of our knowledge, it does not pose foreseeable negative societal or ethical impacts.

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

# A  APPENDIX

*This is the supplementary file for our **ICA** approach and additional experiment results. Contents are organized as follows:*

- *More details of datasets. 1*
- *More details of ICA. 2*
- *Additional comparative results. 3*
- *Training data usage and efficiency. 4*
- *Additional ablation experiments. 5*
- *Additional qualitative results. 6*
- *LLM usage statement. 7*

## A.1  MORE DETAILS OF DATASETS

The disparity in scale, quality, and construction methods of existing public datasets has led to inconsistencies in the training and evaluation protocols for image forgery localization (IFL) methods. These inconsistencies hinder fair and meaningful comparisons across different approaches. To address this issue, we adhere to two widely recognized and rigorous evaluation protocols:

**Protocol-1:** Introduced by MVSS-Net Dong et al. (2022), this protocol trains models exclusively on the CASIAv2 dataset, which consists of 5,123 high-quality tampered images, and directly evaluates them on other datasets without additional fine-tuning. This protocol serves as a robust measure of the model's generalization capabilities.

**Protocol-2:** Used by methods such as PSCC-Net Liu et al. (2022) and CAT-Net Kwon et al. (2022), this protocol trains models on a combined, synthesized dataset. This dataset typically includes CASIAv2 Dong et al. (2013), Fantastic Reality Kniaz et al. (2019), IMD Novozamsky et al. (2020), tampered COCO Lin et al. (2014), tampered RAISE Dang-Nguyen et al. (2015), and other privately synthesized datasets. For example, MVSS-Net Dong et al. (2022) and TruFor Guillaro et al. (2023) are trained on approximately 84,000 and 800,000 synthesized tampered images, respectively.

To overcome these challenges without resorting to large-scale, synthesized pre-training datasets, we propose leveraging a vision foundation model with a prompting mechanism to effectively address IFL tasks. In addition, for localizing tampering in Artificial Intelligence-Generated Content (AIGC) images, we include the CoCoGlide dataset in our analysis. CoCoGlide, created by Guillaro et al. (2023), is a diffusion-based dataset comprising 512 images generated from the COCO 2017 validation set using the GLIDE diffusion model. More visualizations are provided in Figure 4.

### A.1.1  TRAINING AND TESTING PROTOCOLS

To ensure comprehensive and fair comparisons with existing methods, we implement both cross-dataset and homologous-dataset evaluation strategies:

1. **Cross-Dataset Evaluation (Protocol-1):** Following MVSS-Net Dong et al. (2022); Ma et al. (2023); Kong et al. (2025), we train our model exclusively on the CASIAv2 dataset and evaluate its performance on nine additional datasets: CASIAv1+ Dong et al. (2013), NIST16 Guan et al. (2019), Columbia Hsu & Chang (2006), IMD Novozamsky et al. (2020), COVER Wen et al. (2016), DSO-1 Carvalho et al. (2015), DEF-12K Mahfoudi et al. (2019), In-Wild Huh et al. (2018), and Korus Korus & Huang (2016). The detailed information for each dataset can be found in Table 5. This protocol provides a clear benchmark for evaluating the generalization capabilities of the model.

2. **Homologous-Dataset Evaluation (Protocol-2):** Unlike existing methods, which often rely on extensively synthesized datasets, we limit our training to the benchmark training splits of four datasets, avoiding any additional synthesized data. This process, referred to as *benchmark training*, ensures a fairer comparison by mitigating the influence of synthesized pre-training. For homologous experiments, following Wu et al. (2019); Zhou et al. (2023),

Table 5: Detailed information about the selected public datasets. Splicing, Copy-move, Inpainting, and Others are shortened as Cpmv., Spli., Inpa. and Ot., respectively.

| Dataset | Type | | Manipulation Type | | | |
|---|---|---|---|---|---|---|
| | Authentic | Fake | Spli. | Cpmv. | Inpa. | Ot. |
| CASIAv2 Dong et al. (2013) | 7,491 | 5,063 | 3,235 | 1,828 | - | - |
| CASIAv1+ Dong et al. (2013) | 800 | 920 | 459 | 461 | - | - |
| Columbia Hsu & Chang (2006) | 183 | 180 | 180 | - | - | - |
| COVER Wen et al. (2016) | 100 | 100 | - | 100 | - | - |
| IMD Novozamsky et al. (2020) | 414 | 2,010 | - | - | - | 2,010 |
| NIST16 Guan et al. (2019) | - | 564 | 288 | 68 | 208 | - |
| DSO-1 Carvalho et al. (2015) | 100 | 100 | 100 | - | - | - |
| DEF-12K Mahfoudi et al. (2019) | 6,000 | 6,000 | 2,000 | 2,000 | 2,000 | - |
| In-Wild Huh et al. (2018) | - | 201 | 201 | - | - | - |
| Korus Korus & Huang (2016) | 220 | 220 | - | - | - | 220 |
| CoCoGlide Guillaro et al. (2023) | - | 512 | - | - | - | 512 |

our model uses only four benchmark datasets for training and evaluation without incorporating additional synthetic datasets. This benchmark training method only utilizes four benchmark datasets: CASIA Dong et al. (2013), NIST16 Guan et al. (2019), Columbia Hsu & Chang (2006), and COVER Wen et al. (2016). The training and testing splits follow the widely accepted practices in Wu et al. (2019); Zhou et al. (2023). The datasets are divided into training and testing subsets as described in Zhou et al. (2023); Wu et al. (2019), as follows:

- NIST16: 414 images for training, 150 for testing.
- Columbia: 130 images for training, 50 for testing.
- COVER: 75 images for training, 25 for testing.
- CASIAv1+: CASIAv2 for training.

To simulate real-world tampering artifacts and visual degradation, following Ma et al. (2024); Dong et al. (2022); Li et al. (2024b), common data augmentation techniques are applied across all methods, including flipping, blurring, compression, and various naive manipulations such as copy-moving or inpainting rectangular areas within a single image implemented using OpenCV.

**Evaluation Metric.** We evaluate the performance of our model in the task of image forgery localization, similar to previous works Kong et al. (2025); Zhou et al. (2023); Ma et al. (2024). We report the Area Under Curve (AUC) and pixel-level F1 score using the fixed 0.5 thresholds. The average F1 and AUC values of each test dataset are reported as the statistical performance of forgery localization algorithms.

## A.2 MORE DETAILS OF **ICA**

In this study, we employ two distinct Vision Foundation Model (VFM) architectures to evaluate the performance of **ICA**: the single-scale plain Vision Transformer (ViT) Dosovitskiy et al. (2021) and the hierarchical Vision Transformer (SegFormer) Xie et al. (2021). These architectures were chosen to explore the effectiveness of our method across both plain and hierarchical transformer-based designs.

For the plain ViT, we utilize the pre-trained ViT-Base/16 model as the backbone. Since ViT is originally designed for classification tasks, we replace its decoder with a more suitable one for our application. Specifically, we adopt the decoder structure from SETR Zheng et al. (2021), which combines a plain ViT backbone with a progressive upsampling convolutional network as the decoder. This combination facilitates effective feature decoding for pixel-level prediction tasks.

For the hierarchical architecture, we follow the original SegFormer Xie et al. (2021) design, utilizing the MiT-B4 backbone. MiT-B4 is a hierarchical transformer backbone with four stages, enabling multi-scale feature extraction and better handling spatial hierarchies in manipulated images.

Consistent with prior studies Dong et al. (2022); Kong et al. (2025); Zhou et al. (2023), we resize input images to $512 \times 512$ for both training and inference phases. If the original image dimensions

Table 6: Comparison of image forgery localization performance across different models and datasets. The Pixel-level IoU scores (%) (with a fixed threshold: 0.5) are reported. The best scores are highlighted in bold and quoted results marked by '*'.

| Methods AVG | Publication | NIST16 | Columbia | CASIAv1+ | IMD | COVER | DSO-1 | DEF-12K | In-Wild | Korus |
|---|---|---|---|---|---|---|---|---|---|---|
| *Full model training or tuning methods.* | | | | | | | | | | |
| FCN Long et al. (2015) | 11.4 | 17.7 | 36.7 | 15.8 | 11.7 | 4.3 | 8.9 | 14.0 | 8.9 | 14.4 |
| U-Net Ronneberger et al. (2015) | 12.8 | 9.7 | 20.4 | 10.5 | 7.2 | 8.2 | 3.1 | 12.1 | 8.2 | 10.2 |
| DeepLabv3 Chen et al. (2017) | 19.1 | 35.3 | 36.1 | 15.9 | 10.6 | 11.2 | 5.0 | 16.2 | 8.4 | 17.5 |
| MFCN* Salloum et al. (2018) | 19.3 | 12.3 | 29.1 | 12.4 | 10.0 | 10.3 | 5.0 | 11.2 | 8.3 | 13.1 |
| RRU-Net Bi et al. (2019) | 15.6 | 19.6 | 24.4 | 11.9 | 5.7 | 5.7 | 2.4 | 13.1 | 6.8 | 11.7 |
| ManTra-Net Wu et al. (2019) | 9.8 | 30.1 | 11.1 | 9.8 | 13.9 | 15.3 | 3.9 | 20.1 | 6.1 | 13.3 |
| HPFCN* Li & Huang (2019) | 12.6 | 7.6 | 13.7 | 7.6 | 7.0 | 5.4 | 2.6 | 8.4 | 6.4 | 7.92 |
| H-LSTM* Bappy et al. (2019) | 27.6 | 9.0 | 10.1 | 13.1 | 10.8 | 8.4 | 3.7 | 10.6 | 9.4 | 11.4 |
| SPAN Hu et al. (2020) | 15.6 | 39.0 | 11.2 | 10.0 | 10.5 | 4.9 | 2.4 | 13.2 | 5.5 | 12.5 |
| ViT-B Dosovitskiy et al. (2021) | 19.7 | 16.4 | 23.2 | 19.2 | 10.1 | 12.1 | 4.5 | 15.2 | 13.0 | 14.8 |
| Swin-ViT Liu et al. (2021) | 16.7 | 29.7 | 35.6 | 24.3 | 12.4 | 13.2 | 12.9 | 21.4 | 10.3 | 19.6 |
| SegFormer-B4 Xie et al. (2021) | 17.2 | 24.2 | 36.5 | 16.7 | 7.0 | 10.4 | 5.0 | 16.2 | 8.2 | 15.7 |
| PSCC-Net Liu et al. (2022) | 10.8 | 36.0 | 23.2 | 12.0 | 13.0 | 18.5 | 4.2 | 19.3 | 6.6 | 16.0 |
| MVSS-Net++ Dong et al. (2022) | 23.9 | 57.3 | 39.7 | 20.0 | 38.4 | 18.8 | 7.6 | 21.9 | 7.5 | 26.1 |
| CAT-Net Kwon et al. (2022) | 6.2 | 14.0 | 16.5 | 18.3 | 14.1 | 11.0 | 15.2 | 14.4 | 4.9 | 12.7 |
| TruFor* Guillaro et al. (2023) | 21.2 | 78.1 | 48.1 | 29.7 | 21.5 | 15.9 | 12.1 | 30.3 | 9.5 | 29.6 |
| MPC* Lou et al. (2024) | 23.1 | 61.7 | 41.2 | 40.1 | 30.2 | 28.8 | 17.5 | 34.9 | 19.1 | 33.0 |
| IML-ViT Ma et al. (2023) | 25.4 | 68.7 | 64.8 | 35.6 | 37.2 | 4.6 | 24.6 | 12.7 | 2.9 | 30.7 |
| PIM* Kong et al. (2025) | 22.5 | 60.4 | 51.2 | 34.0 | 18.8 | 19.4 | 13.3 | 33.8 | 18.2 | 30.2 |
| *Parameter-efficient tuning methods.* | | | | | | | | | | |
| VPT-Deep Jia et al. (2022) | 18.8 | 35.8 | 41.8 | 24.0 | 5.4 | 1.1 | 13.6 | 14.5 | 8.3 | 18.1 |
| AdaptFormer Chen et al. (2022) | 20.1 | 71.1 | 46.8 | 29.2 | 16.2 | 3.9 | 15.6 | 21.3 | 9.2 | 25.9 |
| EVPv1 Liu et al. (2023a) | 25.7 | 56.1 | 48.8 | 28.9 | 13.3 | 3.3 | 13.9 | 23.6 | 10.4 | 24.9 |
| EVPv2 Liu et al. (2023b) | 23.7 | 51.9 | 49.7 | 30.9 | 16.9 | 10.7 | 14.0 | 23.8 | 12.6 | 26.0 |
| ICA (Ours) | **28.9** | **76.5** | **71.0** | **35.6** | **28.5** | **14.6** | **19.1** | **27.8** | **15.6** | **35.3** |

Table 7: Comparison with state-of-the-art parameter-efficient tuning methods on single-scale Plain ViT (SETR Zheng et al. (2021)). The AUC and F1 scores are reported using both the best and fixed thresholds, with the highest scores highlighted in bold.

| Methods | COVER | | | CASIAv1+ | | | NIST16 | | | Columbia | | |
|---|---|---|---|---|---|---|---|---|---|---|---|---|
| | AUC | Best F1 | Fixed F1 | AUC | Best F1 | Fixed F1 | AUC | Best F1 | Fixed F1 | AUC | Best F1 | Fixed F1 |
| Full-tuning | 0.589 | 0.277 | 0.070 | 0.661 | 0.306 | 0.111 | 0.617 | 0.254 | 0.095 | 0.589 | 0.481 | 0.082 |
| Only Decoder | 0.551 | 0.263 | 0.085 | 0.671 | 0.340 | 0.175 | 0.640 | 0.258 | 0.098 | 0.592 | 0.477 | 0.109 |
| VPT-Deep Jia et al. (2022) | 0.560 | 0.259 | 0.091 | 0.681 | 0.338 | 0.196 | 0.645 | 0.256 | 0.121 | 0.579 | 0.475 | 0.178 |
| AdaptFormer Chen et al. (2022) | 0.577 | 0.273 | 0.086 | 0.684 | 0.339 | 0.200 | 0.647 | 0.276 | 0.108 | 0.624 | 0.498 | 0.094 |
| EVPv1 Liu et al. (2023a) | 0.607 | 0.279 | 0.118 | 0.708 | 0.350 | 0.195 | 0.631 | 0.261 | 0.126 | 0.600 | 0.485 | 0.142 |
| EVPv2 Liu et al. (2023b) | 0.585 | 0.276 | 0.110 | 0.700 | 0.339 | 0.168 | 0.645 | 0.266 | 0.131 | 0.572 | 0.468 | 0.155 |
| ICA (Ours) | **0.708** | **0.350** | **0.200** | **0.714** | **0.364** | **0.215** | **0.661** | **0.297** | **0.168** | 0.619 | **0.502** | **0.207** |

differ from this resolution, we uniformly resize them to ensure compatibility. For ablation studies, we use a lower resolution of $256 \times 256$ to improve computational efficiency. Specifically, the single-scale ViT (SETR) experiments are conducted at this resolution with the following parameter settings for fair comparisons with parameter-efficient tuning methods: 50 prompt tokens for VPT, a middle dimension of 24 for AdaptMLP, a scaling factor $r = 6$ for EVPv1, and $r = 32$ for EVPv2.

For model initialization, the plain ViT (SETR) experiments utilize the ViT-Base model pre-trained on the ImageNet21k dataset Deng et al. (2009) and the hierarchical SegFormer experiments use the MiT-B4 pre-trained on the ImageNet-1k dataset Deng et al. (2009). The localization decoder is initialized using Xavier initialization to ensure a fair comparison.

## A.3 ADDITIONAL COMPARATIVE RESULTS

Tables 6 present the pixel-level localization performance of various SoTA methods on nine unseen datasets using IoU metrics. The methods are divided into two categories: full model training methods (upper section) and parameter-efficient tuning methods (lower section). Among the full model training methods, traditional approaches such as FCN and DeepLabv3 show limited performance, while more advanced methods like IML-ViT achieve a higher average score (AVG) of 33.0, demonstrating improved cross-dataset generalization. However, these gains come at the cost of significant computational resources due to full model retraining or fine-tuning.

For parameter-efficient tuning methods, our proposed ICA outperforms all baselines, achieving the highest scores on six of nine datasets (NIST16, Columbia, CASIAv1+, IMD, COVER, and DEF-12K) and an overall AVG of 35.3, significantly surpassing EVPv2 (26.6). Notably, ICA achieves

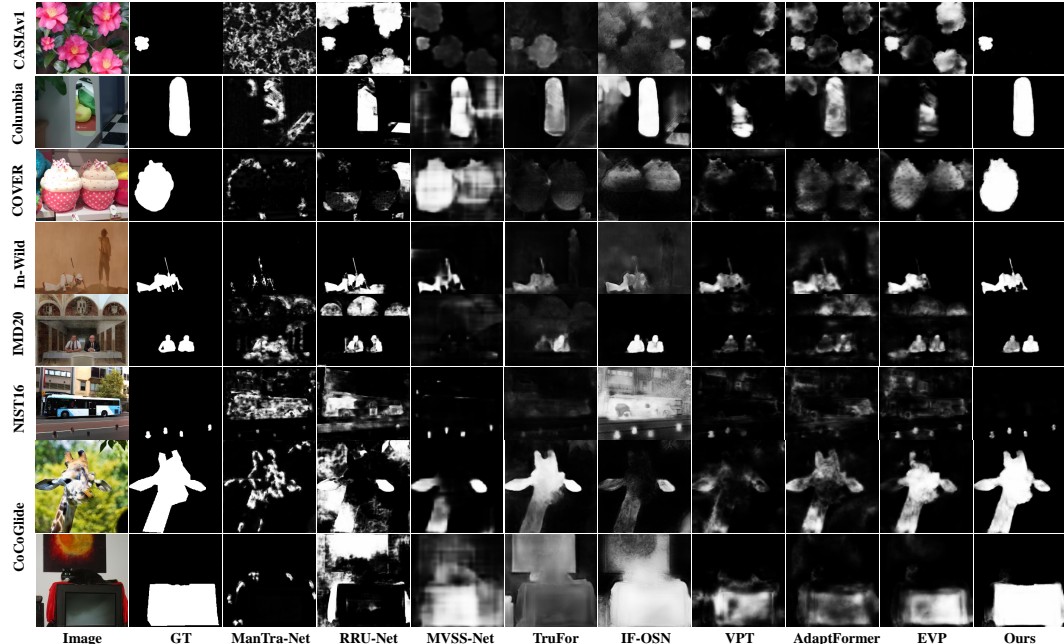

Figure 4: Qualitative comparison between our ICA model and eight full-tuning and different prompting models on seven datasets. The columns, from left to right, present the input images, ground truth (GT) masks, and the predicted masks generated by ManTra-Net Wu et al. (2019), RRU-Net Bi et al. (2019), MVSS-Net Dong et al. (2022), TruFor Guillaro et al. (2023), IF-OSN Wu et al. (2022), VPT Jia et al. (2022), AdaptFormer Chen et al. (2022), EVP Liu et al. (2023a), and our proposed ICA model. Detailed differences are more discernible upon zooming in.

exceptional scores of 76.5 and 71.0 on the Columbia and CASIAv1+ datasets, respectively, highlighting its robustness in diverse forgery localization scenarios. While methods like AdaptFormer and EVPv2 perform well on individual datasets (e.g., Columbia or CASIAv1+), their inconsistent performance across other datasets results in lower average scores.

Overall, ICA demonstrates superior cross-dataset performance and stability. Compared to both full model training and parameter-efficient methods, it strikes an effective balance between efficiency and accuracy, making it well-suited for forgery localization tasks under computational constraints.

To further enhance our approach, we adopt the plain ViT architecture as the backbone for our visual feature model (VFM), detailed in Section A.2. Table 7 compares ICA with state-of-the-art parameter-efficient tuning methods using single-scale Plain ViT (SETR). ICA consistently achieves the highest AUC and F1 scores under both best and fixed thresholds, demonstrating its robustness and adaptability even with a simplified backbone like Plain ViT. This performance can be attributed to the explicit and implicit prompting mechanisms, which integrate knowledge from multiple forensic domains and effectively leverage and transfer the frozen VFM knowledge to efficiently address forgery localization tasks. Furthermore, the proposed method performs well across different architectures, including plain ViT and SegFormer, underscoring its generality and effectiveness in various settings.

Table 8 presents quantitative results to assess the effectiveness of our method in AIGC tampering localization. CNN-based methods (MVSS-Net) struggle with diverse tampering patterns, achieving only 15.0% F1. Transformer-based models (IML-ViT) performance (20.1% F1) by modeling global relationships but remain limited in capturing diverse forgery patterns. In contrast, our ICA strategy significantly outperforms all existing methods, achieving 33.9% F1 and 27.6% IoU, effectively capturing both low-level forensic traces and high-level semantic inconsistencies and improving generalization.

Table 8: Quantitative comparison on AI-generated content detection using the CoCoGlide dataset.

| Method | MVSS-Net | VPT | AdaptFormer | EVPv1 | IML-ViT | Ours |
|---|---|---|---|---|---|---|
| Pixel-level F1 (%) | 15.0 | 15.6 | 17.4 | 16.2 | 20.1 | 33.9 |
| IoU (%) | 10.7 | 10.5 | 12.3 | 11.7 | 15.8 | 27.6 |

Table 9: Comparison of models in terms of training data usage, parameter size, computational cost, and AUC performance. Our ICA achieves state-of-the-art AUC with minimal training data and competitive efficiency.

| Model | Training Data (K) | Params (M) | FLOPs (G) | AUC |
|---|---|---|---|---|
| MVSS-Net | 96.60 | 142.79 | 327.14 | 0.716 |
| ObjectFormer | - | 257.97 | 402.80 | 0.884 |
| TruFor | 900.25 | 262.05 | 519.91 | 0.927 |
| UnionFormer | 832.50 | 210.63 | 392.82 | 0.929 |
| EVP | 5.12 | 64.54 | 86.77 | 0.898 |
| Adaptformer | 5.12 | 64.05 | 85.48 | 0.886 |
| ICA (Ours) | 5.12 | 67.51 | 121.28 | 0.963 |

A.4 TRAINING DATA USAGE AND EFFICIENCY

Table 9 provides a comparative analysis of various models (MVSS-Net Dong et al. (2022), ObjectFormer Wang et al. (2022), TruFor Guillaro et al. (2023), EVP Liu et al. (2023a), UnionFormer Li et al. (2024b), Adaptformer Chen et al. (2022)) in terms of training data requirements, parameter size, computational cost (measured in FLOPs), and performance (measured by average AUC across five datasets). ICA achieves the highest AUC of 0.963, significantly outperforming all baselines while using only 5.12K training images. In contrast, full model training methods such as TruFor and UnionFormer require up to 900.25K and 832.50K samples, respectively, while achieving lower AUCs of 0.927 and 0.929. This underscores ICA's ability to achieve superior performance with just 0.57% of the training data required by TruFor.

Among parameter-efficient methods, EVPv1 and AdaptFormer have comparable parameter sizes (64.54M and 64.05M) but achieve lower AUCs (0.898 and 0.886). While ICA has slightly higher FLOPs (121.28G compared to 86.77G for EVPv1), the AUC improvement of up to 7.2% justifies the marginal increase in computational cost.

These results demonstrate that ICA achieves an optimal balance of minimal training data usage, compact model size, and superior accuracy, making it highly suitable for practical forgery localization tasks where computational resources and training data are constrained.

A.5 ADDITIONAL ABLATION EXPERIMENTS

To evaluate the contributions of different stages in ICA, we performed ablation studies by varying the tunable stages within the SegFormer backbone. The SegFormer encoder comprises four hierarchical stages, each responsible for extracting features at different scales. In our experiments, the inclusion of tunable prompting at a specific stage is denoted as $\text{Stage}_x$, where $x$ corresponds to one or more stages (e.g., $x = 1, 2, 3, 4$).

In Table 10, the results reveal that performance improves consistently across datasets as the number of tunable stages increases, demonstrating the effectiveness of our prompting method. Notably, the most significant performance gain occurs when transitioning from $\text{Stage}_{1,2}$ to $\text{Stage}_{1,2,3}$, indicating that tuning $\text{Stage}_3$ contributes the most to overall improvement. It is important to note that in SegFormer-B4, the number of transformer blocks in each stage is 3, 8, 27, and 3, respectively. This observation suggests that the performance of ICA is positively correlated with the number of prompted transformer blocks, particularly when intermediate stages with a higher number of blocks (e.g., $\text{Stage}_3$) are included in the tuning process.

Table 10: Ablation studies across different tuning stages of the SegFormer backbone on multiple datasets. Metrics include trainable parameters and $F_1$ scores.

| Tuning Stage | Trainable Param. | CASIAv1+ | COVER | Columbia |
|---|---|---|---|---|
| Stage$_1$ | 4.84M | 0.447 | 0.157 | 0.303 |
| Stage$_{1,2}$ | 4.91M | 0.568 | 0.155 | 0.300 |
| Stage$_{1,2,3}$ | 6.21M | 0.667 | 0.229 | 0.437 |
| Stage$_{1,2,3,4}$ | 6.90M | 0.678 | 0.236 | 0.440 |

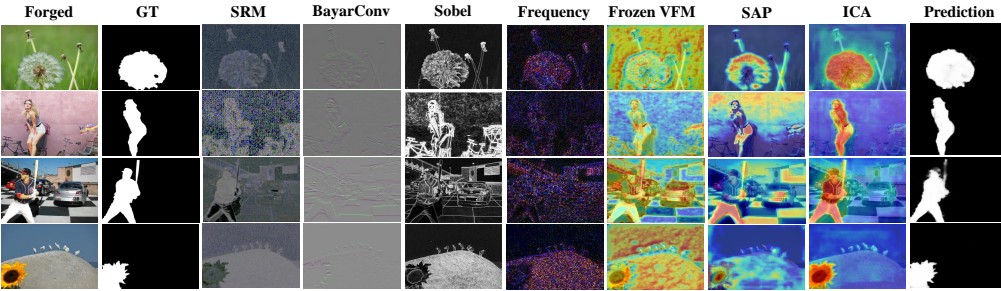

Figure 5: Visualization of diverse features. From left to right: image, GT mask, noise-aware prompts (SRM, BayarConv, Sobel), frequency-aware prompts, and CAMs of feature maps from frozen VFM, SAP, and ICA framework, followed by the final prediction.

## A.6 ADDITIONAL QUALITATIVE RESULTS

Figure 4 provides a qualitative comparison between our proposed ICA model and eight baseline models across seven datasets. Each row corresponds to a specific dataset, while columns present input images, ground truth (GT) masks, and the predicted masks generated by competing methods, including ManTra-Net Wu et al. (2019), RRU-Net Bi et al. (2019), MVSS-Net Dong et al. (2022), TruFor Guillaro et al. (2023), IF-OSN Wu et al. (2022), VPT Jia et al. (2022), AdaptFormer Chen et al. (2022), EVP Liu et al. (2023a), and ICA.

Our ICA model demonstrates superior localization accuracy, particularly in complex scenarios with subtle or irregular forgery patterns. As shown in Figure 4, the predicted masks produced by ICA are visually closer to the ground truth, exhibiting finer granularity and better boundary preservation compared to other methods. On the CASIA1 and COVER datasets, ICA effectively highlights tampered regions with minimal noise, outperforming traditional full-tuning models such as MVSS-Net and IF-OSN. On datasets with more challenging real-world scenarios, such as In-Wild and IMD, ICA captures subtle forgery artifacts that other models often fail to detect, as evidenced by its cleaner and more accurate masks. Furthermore, compared to parameter-efficient models like VPT and Adapt-Former, ICA generates more detailed and precise predictions, demonstrating its superiority.

Besides, Figure 5 illustrates the noise-aware, frequency-aware features, and prompt-guided heatmaps in ICA. In Columns 1–6, some images appear natural in RGB view but reveal tampered artifacts when analyzed in noise or frequency domains. Column 7 presents the heatmap from a frozen VFM, while Column 8 demonstrates how the model, guided by SAP, focuses on forgery regions. Column 9 highlights ICA's ability to locate tampered areas more precisely, leveraging forgery cues for accurate predictions. More qualitative results are provided in the *Supplementary Material*.

These qualitative results are consistent with the quantitative metrics, further underscoring the robustness and effectiveness of ICA in diverse forgery localization tasks.

## A.7 LLM USAGE STATEMENT

In accordance with the ICLR policy on the use of large language models (LLMs), we disclose their role in the preparation of this paper. LLMs (e.g., ChatGPT4o) were employed solely as a general-purpose writing assistant. Specifically, they were used to polish grammar, improve clarity, and suggest alternative phrasings for certain sections of the manuscript. All research ideas, methodolog-

ical designs, experiments, and analyses were entirely conceived and conducted by the authors. The LLMs did not contribute to research ideation, experimental design.

