# OpenReview forum: "In-Context Alignment: Resolving Representation Conflict for Parameter-Efficient Forgery Detection in Vision Models"
_ICLR.cc/2026/Conference — Submitted to ICLR 2026_

### Official Review · Reviewer_cvu9 · 2025-10-29

**Soundness:** 3
**Presentation:** 3
**Contribution:** 2
**Rating:** 4
**Confidence:** 4

**Summary:**

The paper proposes a new transformer based architecture for image forgery localization and detection. They use an existing pretrained vision transformer and augment it using adapter layers termed PAP and SAP. The additional layers are aimed at transforming the intermediate token representations with noise and semantic features necessary for forgery detection.

The contributions of the paper include
- A novel approach to incorporate noise level features to align the token representations to the goal without training or finetuning the base model.
- A mixture of experts approach to combine multi-domain features to enhance token representation
- Empirical evidence to support the claims

**Strengths:**

- The objective of reducing the distance between latent representation of content tokens and noise tokens is a good approach to tackling IFLD task. One of the major challenges of this domain is how to utilize existing pretrained models effectively without representation collapse.
- The paper proposes a physical aware module that combines different noise and latent features of the input using mixture of experts fusion. They also handle the problem of expert balancing by tuning the loss function.
- The authors focus on parameter efficiency and minimizing complexity which is clearly explained in the section on complexity.
- The authors perform extensive experimental evaluation of various datasets and benchmarks, which show state of the art results
- The paper is well written and clearly explains what steps were followed to achieve the results

**Weaknesses:**

- The paper does not include any analysis of how each feature / module effects the final performance. For example, how much do the noise feature, or frequency features, or the semantic features have an effect on the final prediction?
- The paper does not have any new major contribution to the task of image forensics in general. Almost all works in this field use the same standard combination of noise filters, frequency filters, and combine them with image features for the final detection. Although it might be interesting to see how the feature fusion strategy might work in different domains or a different context, in terms of IFLD, this work does not propose any major new contribution.

**Questions:**

- What is the motivation of using moe fusion? The goal of using mixture of experts is to reduce inference complexity by reducing the number of active parameters. To this end moe models only activate certain experts for some specific token. But the current framework focuses on all experts being equally weighted so that all the different noise features are used by the model. If this is the case why not just concatenate or combine the features in a different way? The usefulness of using moe needs more justification.
- The model was trained only using CASIAv2 which  does not have any inpainting or neural forgery samples. However the model still shows high accuracy on datasets like Nist16 or DEF12k which have a large portion of inpainting samples. I request the authors to perform forgery specific evaluation to better understand the models capabilities.
- The paper does not assess the limitations or failure cases of the model.

---

### Official Review · Reviewer_F8mB · 2025-10-30

**Soundness:** 3
**Presentation:** 3
**Contribution:** 2
**Rating:** 6
**Confidence:** 3

**Summary:**

This paper proposes In-Context Alignment (ICA), a parameter-efficient framework for image forgery localization (IFL). ICA resolves the representation conflict in Vision Foundation Models (VFMs) by introducing two complementary prompting modules: the Physical-Aware Prompter (PAP), which amplifies low-level forensic cues, and the Semantic-Aware Prompter (SAP), which exposes high-level semantic inconsistencies. Built upon a frozen SegFormer backbone, ICA achieves robust localization with minimal trainable parameters.

**Strengths:**

1. The paper clearly identifies and analyzes the limitations of Vision Foundation Models (VFMs) in image forgery localization, and designs corresponding modules to address deficiencies in both physical and semantic perception. The motivation is well-grounded, the methodology is logically derived from the problem analysis, and the overall framework achieves effective adaptation of VFMs with high efficiency.
2. The paper presents clear and well-structured descriptions, with precise mathematical formulations and theoretically sound derivations that enhance the overall rigor and credibility of the proposed framework.
3. The paper conducts extensive experiments under various evaluation settings, verifying the effectiveness and robustness of the proposed method.

**Weaknesses:**

1. The datasets used in the experiments are outdated. Given the rapid progress of diffusion-based image editing in recent years and the emergence of new datasets such as RealEdit, incorporating more up-to-date benchmarks to enhance the practical relevance of the proposed IFDL framework in real-world scenarios.
2. The design of the Embedding Expert within the PAP module lacks sufficient clarification. From its structural perspective, it appears to model features at a semantic level rather than a physical level.
3. The ablation study is insufficient. The paper does not provide experimental evidence demonstrating the effectiveness of the PAP and SAP modules or their subcomponents. Moreover, in the Optimization section, the choice of weighting parameters for different loss terms lacks ablation analysis to justify their settings.
4. Since efficiency is a key advantage of the proposed method, it is recommended to include additional metrics such as trainable parameters in Table 1 to provide a more comprehensive and quantitative comparison of efficiency.

**Questions:**

Please  see the weakness.

---

### Official Review · Reviewer_DnqJ · 2025-10-31

**Soundness:** 2
**Presentation:** 2
**Contribution:** 2
**Rating:** 2
**Confidence:** 4

**Summary:**

This paper proposes In-Context Alignment (ICA), a parameter-efficient framework for image forgery localization that adapts vision foundation models using two modules. The authors argue that vision foundation models have a representation conflict between content-oriented and forgery-sensitive features. To combat this the authors propose two modules: a Physical-Aware prompter for low-level inconsistencies using a MoE design and a Semantic-Aware prompter for high-level inconsistencies. The foundation model backbone remains frozen, allowing for parameter-efficient updates. ICA is evaluated across a wide range of baselines under multiple training regimes and robustness experiments.

**Strengths:**

- The representation conflict is a meaningful concept to introduce.
- The proposed method is parameter-efficient whilst attaining strong generalization.
- The paper contains an extensive list of performance and robustness experiments.

**Weaknesses:**

- The representation conflict is only explored in terms of ICA for forgery detection and localization, which is a very indirect approach. It could have been explored more thoroughly by using full-finetuning and LoRA/probing as baselines for foundation models. This would show the magnitude of the conflict.
- The evaluation is performed on outdated baselines, while recently published IFL papers such as FakeShield [1] and SAFIRE [2] report higher F1 scores on many of the datasets, and focus on more recent datasets.
- The evaluation is performed on outdated datasets that focus on legacy edit methods such as splicing and copy-move. The AIGC dataset, CoCoGLIDE, is outdated as diffusion models have advanced significantly.  Even then, the author’s CoCoGLIDE result substantially underperforms the state-of-the-art: 33.9 F1 vs 63.5 F1 of SAFIRE  and 52.3 F1 of TruFor, which are left out of the comparison. Therefore, we highly doubt the relevance of the proposed method in the context of modern edits such as those in SIDA [3].
- In the ablation, the model without the shared MLP performs best; we are unsure whether this is the chosen model.
- The robustness evaluation is done only on images from NIST16, limiting the generalizability of the results. Further, it is unclear to us why JPEG compression with a quality factor of 100 is a valid test of robustness.
- We identified several inconsistencies in Table 1. Specifically, we highlight two inconsistencies. In general, however, the authors report lower scores in this paper than in others that implement the same baselines.
  - TruFor reports pixel F1 scores for (NIST16, Columbia, CASIAv1, Coverage) are (39.9,85.9,73.7,60), respectively. In the author’s paper, the scores for TruFor on those datasets are (26.8, 82.9, 53.2, 28) . This possibly means TruFor outperforms ICA but is misreported.
  - Similarly, quoting the scores from the CAT-Net paper, we see a pixel-level localization IoU on NIST16 and Columbia of 68.41, 83.05, whereas the authors report 10.2 and 20.6, respectively.


[1] Xu, Z., Zhang, X., et al. (2025). FakeShield: Explainable image forgery detection and localization via multi-modal large language models. In Proceedings of the International Conference on Learning Representations (ICLR 2025).​

[2] Kwon, M.-J., Lee, W., et al. (2025). SAFIRE: Segment any forged image region. In Proceedings of the AAAI Conference on Artificial Intelligence (AAAI 2025).​

[3] Huang, Z., Hu, J., Li, X., He, Y., Zhao, X., Peng, B., Wu, B., Huang, X., & Cheng, G. (2025). SIDA: Social media image deepfake detection, localization and explanation with large multimodal model. In Proceedings of the IEEE/CVF Conference on Computer Vision and Pattern Recognition (CVPR 2025).

**Questions:**

- Why are recent image forgery detection and localization models not included as baselines?
- Why is AIGC capability only evaluated on CoCoGLIDE (2023) and only to 5 baselines?
- Is the model “w/o shared MLP” the chosen model in the ablation Table 4?
- Why is robustness evaluated only on NIST16, and what is the rationale for using JPEG quality factor 100?
- Why are the TruFor and CAT-Net scores significantly lower in the authors’ paper vs. other papers (TruFor paper, FakeShield [1], CAT-NET, SAFIRE [2])?

[1] Xu, Z., Zhang, X., et al. (2025). FakeShield: Explainable image forgery detection and localization via multi-modal large language models. In Proceedings of the International Conference on Learning Representations (ICLR 2025).​

[2] Kwon, M.-J., Lee, W., et al. (2025). SAFIRE: Segment any forged image region. In Proceedings of the AAAI Conference on Artificial Intelligence (AAAI 2025).​

---

### Official Review · Reviewer_AaeU · 2025-10-31

**Soundness:** 2
**Presentation:** 3
**Contribution:** 1
**Rating:** 2
**Confidence:** 5

**Summary:**

The authors propose In-Context Alignment, a parameter-efficient framework that reformulates forgery localization as a visual in-context learning task. ICA introduces two complementary prompting modules within frozen VFMs Physical-Aware Prompter and Semantic-Aware Prompter. The designed modules jointly enhance low-level forensic signals and encourage the model to highlight semantic inconsistencies in high-level features. The intra-domain, cross-domain, and ablation experiments well supported the effectiveness of the proposed method.

**Strengths:**

The paper is well-organized, and the experimental results are comprehensive and convincingly support the proposed method.

**Weaknesses:**

- Previous MLLM-based forensics models primarily focus on the textual explainability of model predictions. In contrast, this work only provides forgery mask predictions without any textual reasoning. Moreover, the cross-dataset generalization performance is inferior to previous methods on most unseen datasets.The motivation of using vision foundation model is unclear.

- The manuscript claims that the feature representation  C can be decomposed as  C≈C_physical+C_semantic, and that the proposed method achieves a tighter bound. However, these claims lack rigorous theoretical analysis or formal derivation, which weakens the technical credibility of the argument.

- More comprehensive metrics such as F1 and IoU scores should be reported in the homologous-dataset evaluations to allow a more complete and interpretable assessment of model performance.

- The adopted data augmentation types overlap substantially with the perturbations used in robustness evaluation. This overlap may inflate the perceived robustness. In real-world applications, models often face more complex and unseen perturbations, which are not reflected in the current experimental setup.

- The claim that low-level artifact cues and high-level semantic cues are orthogonal components is not sufficiently justified or empirically validated. A deeper theoretical or statistical analysis would be needed to support this assumption.

- The role and quantitative contribution of the designated Embedding Expert module remain unclear. More justifications should be included to clarify its impact on the final performance.

The authors have overstated their method’s superiority and contributions in several parts of the manuscript.

- In Tables 1 and 6, the captions claim that the best-performing results are highlighted in bold. However, the proposed method does not achieve the best performance on multiple datasets, such as NIST16, Columbia, IMD, COVER, DSO-1, In-the-Wild, and Korus. Despite this, these results are still shown in bold, which is highly misleading and potentially confusing to readers.

- The authors repeatedly emphasize that their approach is efficient due to the use of PEFT. However, no complexity analysis or empirical evidence is provided to substantiate this claim. The paper must include quantitative evaluations of efficiency, such as FLOPs, training speed, GPU memory usage, and inference time.

- The claimed novelty and contributions are overstated. The use of PEFT has already been extensively explored in prior LVLM-based forensics studies, including So-Fake [a], SIDA [b], and FakeShield [c]. The concept of forensics-aware prompt learning for image forgery localization was introduced in ForgeryGPT [d]. The integration of low-level forensic and high-level semantic features within MLLMs has also been addressed in Propose and Rectify [e]. Additionally, the noise extractors adopted here (Bayar, Sobel, and SRM) are nearly identical to those used in [e]. The MoE mechanism has been extensively studied in prior forensics research and cannot be considered novel in this context.

[a] So-Fake: Benchmarking and Explaining Social Media Image Forgery Detection

[b] SIDA: Social Media Image Deepfake Detection, Localization and Explanation with Large Multimodal Model

[c] FakeShield: Explainable Image Forgery Detection and Localization via Multi-modal Large Language Models

[d] ForgeryGPT: Multimodal Large Language Model For Explainable Image Forgery Detection and Localization

[e] Propose and Rectify: A Forensics-Driven MLLM Framework for Image Manipulation Localization

**Questions:**

See weaknesses.

---

### Meta-Review · Area_Chair_djPy · 2026-01-05

**Summary:**

While the reviewers see some merits in the proposal of in-context alignment for forgery detection in vision modes, this submission should not be accepted in its current form due to several fundamental issues, as pointed out by the reviewers, including

- The evaluation datasets and comparative methods are outdated
- As the authors mentioned in the rebuttal, the proposed method shows potential advantage in the regime of few-data settings. However, direct comparisons to advanced detections in the same setting were missing, which limits the contributions of the proposed method.

Overall, this paper requires significant modifications and another round of full review.

**Reviewer Concerns:**

The comments from Reviewer AaeU and Reviewer Dnqj are critical yet unaddressed.

**Reviewer Scores:**

Even if some reviewers tend to increase the scores, Reviewer AaeU and Reviewer Dnqj would not be convinced by the rebuttal.

---

### Decision · Program_Chairs · 2026-01-26

Reject